# Mutation in the Ciliary Protein C2CD3 Reveals Organ-Specific Mechanisms of Hedgehog Signal Transduction in Avian Embryos

**DOI:** 10.3390/jdb9020012

**Published:** 2021-03-25

**Authors:** Evan C. Brooks, Christian Louis Bonatto Paese, Anne H. Carroll, Jaime N. Struve, Nandor Nagy, Samantha A. Brugmann

**Affiliations:** 1Division of Developmental Biology, Cincinnati Children’s Hospital Medical Center, Cincinnati, OH 45229, USA; Evan.brooks@cchmc.org (E.C.B.); christian.bonattopaese@cchmc.org (C.L.B.P.); Anne.Carroll@cchmc.org (A.H.C.); jnstruve3@gmail.com (J.N.S.); 2Division of Plastic Surgery, Cincinnati Children’s Hospital Medical Center, Cincinnati, OH 45229, USA; 3Department of Anatomy, Histology and Embryology, Faculty of Medicine, Semmelweis University, 1094 Budapest, Hungary; nagy.nandor@med.semmelweis-univ.hu; 4Shriners Hospital for Children–Cincinnati, Cincinnati, OH 45229, USA

**Keywords:** primary cilia, ciliopathies, Hedgehog signaling, *talpid*^2^, C2CD3, hindgut, neural tube, craniofacial, limb

## Abstract

Primary cilia are ubiquitous microtubule-based organelles that serve as signaling hubs for numerous developmental pathways, most notably the Hedgehog (Hh) pathway. Defects in the structure or function of primary cilia result in a class of diseases called ciliopathies. It is well known that primary cilia participate in transducing a Hh signal, and as such ciliopathies frequently present with phenotypes indicative of aberrant Hh function. Interestingly, the exact mechanisms of cilia-dependent Hh signaling transduction are unclear as some ciliopathic animal models simultaneously present with gain-of-Hh phenotypes in one organ system and loss-of-Hh phenotypes in another. To better understand how Hh signaling is perturbed across different tissues in ciliopathic conditions, we examined four distinct Hh-dependent signaling centers in the naturally occurring avian ciliopathic mutant *talpid*^2^ (*ta*^2^). In addition to the well-known and previously reported limb and craniofacial malformations, we observed dorsal-ventral patterning defects in the neural tube, and a shortened gastrointestinal tract. Molecular analyses for elements of the Hh pathway revealed that the loss of cilia impact transduction of an Hh signal in a tissue-specific manner at variable levels of the pathway. These studies will provide increased knowledge into how impaired ciliogenesis differentially regulates Hh signaling across tissues and will provide potential avenues for future targeted therapeutic treatments.

## 1. Introduction

Primary cilia are ubiquitous microtubule-based organelles that sense the molecular and mechanical environment of the cell [1]. When there are disruptions in the structure or function of primary cilia, a growing class of heterogenous disorders called ciliopathies arise. Clinical features associated with ciliopathies include phenotypes spanning a number of organ systems such as the eye (retinitis pigmentosa), internal organs (situs inversus, renal cystic disease), and limbs (polydactyly) [2]. Current treatment options for ciliopathies are extremely limited as understanding of ciliary function during development remains incomplete.

In vertebrates, primary cilia are associated with the transduction of numerous signaling pathways, most notably the Hedgehog (Hh) signaling pathway [3]. The Hh pathway has been inextricably linked to the primary cilium as intraflagellar transport (IFT) proteins that participate in the bidirectional transport of particles necessary for ciliary extension are required for Hh signal propagation [3,4,5]. The pathway is activated when a Hh ligand binds to the Patched1 (Ptch1) receptor [6,7,8]. Hh-Ptch1 binding relieves the Ptch1-mediated inhibition of the Hh pathway transducer Smoothened (Smo) [9]. When Smo inhibition is relieved, Smo translocates into the cilium and accumulates within the ciliary axoneme [10,11]. Smo then transduces the Hh signal through the glioma-associated oncogene (Gli) family of transcription factors (Gli1, Gli2, Gli3). Gli1 is a downstream target of the Hh pathway and acts as a potentiator that contributes to a threshold required for pathway activation [12,13,14,15]. Gli2 and Gli3 can function as either activators or repressors of the Hh pathway with Gli2 functioning as the primary activator and Gli3 functioning as the primary repressor [15,16,17,18,19,20]. Hh pathway activation promotes the processing of the Gli transcription factors into their full-length activator isoforms (GliA) that will activate target gene expression [21]. In the absence of an Hh ligand, Smo is repressed by Ptch1 and the Gli transcription factors are proteolytically processed into their truncated repressor isoforms (GliR) that will repress target gene expression. Post-translational processing of Gli2/3 into their GliA and GliR isoforms occurs at the primary cilium and disrupted ciliogenesis impairs the production of GliA and GliR isoforms [1,22,23].

Proper Gli processing in the primary cilium is required for transcription of Hh target genes, including the genes encoding the Hh pathway receptor *Ptch1* and the Hh pathway potentiator *Gli1* [24,25]. Both *Ptch1* and *Gli1* expression are markers of Hh pathway activity as their transcription is dependent upon Hh pathway activation and their expression is adjacent to *Shh* [6,15,26,27]. One of the most important features of the Hh pathway is its inherent self-regulating properties by various pathway components. One such component with a prominent role in Hh pathway regulation is Ptch1, which antagonizes the actions of Hh ligands and keeps the pathway in a constitutive off-state in the absence of a Hh ligand [28]. Additionally, Ptch1 protein represses transcription of the *Ptch1* gene and other Hh target genes by sequestering the Hh signal [9,27,28,29]. Conversely, the *Ptch1* gene is strongly upregulated as a result of Hh pathway activation through inhibition of Ptch1 protein function [24,27,30]. The upregulated expression of the *Ptch1* transcript upon Hh pathway activation and its subsequent repression by the Ptch1 protein demonstrates the presence of a negative feedback loop that balances Hh pathway activity. This self-regulated tempering of the Hh pathway through the Ptch1 negative feedback loop determines the level of Hh target gene expression and overall Hh pathway activity. Thus, *Ptch1* transcript expression is the most reliable indication of Hh pathway activity in both vertebrates and invertebrates [6,31,32,33].

Despite an early understanding that primary cilia were solely required to promote Hh signaling, numerous studies suggested a more complex function in signal transduction. Several ciliopathic animal models present with tissue-specific phenotypes indicative of either a gain-of-Hh or loss-of-Hh function. For instance, murine embryos with mutations in the anterograde IFT component *Intraflagellar transport protein 88* (*Ift88)* and the retrograde IFT component *Dynein cytoplasmic heavy chain 2* (*Dnchc2*) present with both polydactyly, a gain-of-Hh phenotype, and loss of ventral neural progenitors in the neural tube, a loss-of-Hh phenotype [4,5]. When the anterograde IFT component *Kinesin family member 3A* (*Kif3a*) is conditionally deleted from neural crest cells (NCCs), mutant embryos simultaneously present with gain-of-Hh phenotypes including midfacial widening [34,35,36] and loss-of-Hh phenotypes such as micrognathia, aglossia, submandibular gland aplasia, and absent incisors [36,37,38,39,40]. While significant insights have been made into the molecular etiologies of ciliopathies, a comprehensive understanding of how loss of specific ciliary components impact distinct Hh signaling centers, including the limb, the neural tube, the craniofacial complex, and the gastrointestinal (GI) system, is incomplete.

The *talpid*^2^ (*ta*^2^) mutant is a long-utilized, naturally occurring avian mutant that is characterized by limb and craniofacial phenotypes [41,42]. Our previous studies revealed that the *ta*^2^ presentation is the result of a 19 bp deletion in *C2 calcium-dependent domain containing 3* (*C2CD3*) [43], a gene encoding for a distal centriolar protein required for ciliogenesis [44,45]. Mutation in *C2CD3* results in impaired ciliogenesis in the *ta*^2^ and subsequent polydactyly, facial clefting, ectopic archosaurian-like first generation teeth, hypo- or aglossia, and micrognathia [36,43,46,47,48,49,50,51]. While previous studies demonstrated that Hh signaling was disrupted in various tissues of the *ta*^2^ embryo [36,43,50,51,52,53,54], these studies were limited in making causal genotype-phenotype connections because the genetic cause of the *ta*^2^ mutation was unknown and early Hh patterning events occurred prior to the emergence of distinguishing phenotypes. The recent discovery of the *ta*^2^ mutation, coupled with the ease of genotyping, now allows for the reanalysis of Hh signaling events prior to the onset of morphological phenotypes between *ta*^2^ mutant, heterozygous, and control embryos.

In this study, we analyze Hh-dependent signaling centers associated with both gain- and loss-of-function phenotypes in the *ta*^2^ mutant. Interestingly, phenotypic and molecular analyses of *SHH* and *PTCH1* expression suggest that distinct modes of cilia- and Hh-dependent patterning are utilized throughout the embryo.

## 2. Materials and Methods

### 2.1. Avian Embryo Collection, Genotyping, and Tissue Preparation

Fertilized control and *ta*^2^ eggs were supplied from University of California, Davis. Embryos were incubated at 38.8 °C for 3–13 days and then harvested for analysis. All embryos collected were staged according to the Hamburger–Hamilton (HH) staging system [55]. Embryos were genotyped as previously described [43]. For all experiments, 5 control^+/+^ and 5 *ta*^2^ embryos were utilized, unless noted otherwise in the figure legend. HH20 embryos were fixed in 4% paraformaldehyde (PFA) diluted in phosphate buffer solution (PBS) overnight at 4 °C. HH24 hindlimbs were fixed in 4% PFA for 2 h at room temperature. HH29-HH39 hindguts were fixed in 4% PFA for 30 min at room temperature.

### 2.2. Wholemount Skeletal Staining

Wholemount skeletal staining of HH39 limbs was carried out as previously described [46].

### 2.3. RNAscope In Situ Hybridization Assay

For RNAscope [56] on tissue sections, HH20 embryos and HH29-HH39 hindguts were dehydrated in an ethanol (EtOH) series, washed in xylene, embedded in paraffin, and sectioned at 8 μm thickness. Transcripts of *SHH* (Advanced Cell Diagnostics (ACD) 551581), *PTCH1* (ACD 551571), *NKX2.2* (ACD 551551), *OLIG2* (ACD 551561-C2), *IRX3* (ACD 551611-C3), *FGF8* (ACD 868491), and *OTX2* (ACD 902291-C2) were detected using the RNAscope Multiplex Fluorescent V2 kit (ACD, Newark, CA, USA) per manufacturer’s instructions. Signal development for each probe was carried out using either fluorescein (Akoya Biosciences NEL741001KT, Marlborough, MA, USA) or Cyanine 3 (Akoya Biosciences NEL744001KT, Marlborough, MA, USA) diluted 1:500 in RNAscope Multiplex TSA Buffer. Slides were counterstained with 4′,6-diamidino-2-phenylindole (DAPI) (ACD, Newark, CA, USA), mounted with Prolong Gold (Invitrogen P36930, ThermoFisher Scientific, Waltham, MA, USA), and imaged using a Leica DM5000B (Leica Microsystems, Buffalo Grove, IL, USA) upright microscope system. 

For wholemount RNAscope, HH24 hindlimbs were prepared as previously described with slight modifications [57]. Briefly, HH24 embryos were dehydrated in a methanol/PBS with 0.01% Tween graded series and stored in 100% methanol at −20 °C overnight. Embryos were treated with 3% hydrogen peroxide in PBS for 30 min to inactivate endogenous peroxidase activity. Incubation time in RNAscope Protease III was increased to 12 min. Transcripts of *SHH* (ACD 551581) and *PTCH1* (ACD 551571-C2) were detected using the RNAscope Fluorescent Multiplex kit (ACD, Newark, CA, USA) per manufacturer’s instructions. Nuclei were counterstained with RNAscope DAPI. Hindlimbs were cleared in Ce3D+ solution [58] overnight and mounted in 35 mm glass bottom dishes (MatTek P35 G-0-7-C).

### 2.4. RNAscope Puncta Quantification

All RNAscope samples were imaged using a Nikon A1R (Nikon Instruments, Melville, NY, USA) inverted confocal microscope at 60× magnification. Images were loaded into Imaris 9.6.0 (Bitplane, Concord, MA, USA). To count the RNAscope puncta for specific molecular markers, the Spots tool was utilized. For the neural tube markers *NKX2.2*, *OLIG2,* and *IRX3*, the region of expression was segmented from the entire image to only process the number of puncta within the region of interest. The estimated diameter of the puncta was set to 0.5 µm and the background was subtracted. A quality score filter was set to separate the RNAscope puncta from background. Statistical analysis was performed using a two-tailed Student’s *t*-test at the 0.05 significance level with *p* < 0.05 being statistically significant.

### 2.5. Morphometric Measurements

Measurements of *SHH*+ areas in the limb, neural tube, and hindgut; *PTCH1+*, *NKX2.2*+, *OLIG2*+, *IRX3*+ areas in the neural tube; and the total areas of the limb and neural tube were performed in Imaris 9.6.0. For the area measurements for each molecular marker, the region of expression in each tissue was segmented from the entire image. The Surfaces tool was utilized to calculate the area of the region in each tissue and the total areas of the limb and the neural tube. Statistical analysis was performed using a two-tailed Student’s *t*-test at the 0.05 significance level with *p* < 0.05 being statistically significant.

GI organ length measurements were performed in FIJI [59]. The freehand line tool was utilized to trace the length of the organ of interest. Statistical analysis was performed using a two-tailed Student’s *t*-test at the 0.05 significance level with *p* < 0.05 being statistically significant.

### 2.6. Immunohistochemistry

For P75 and VERSICAN detection, hindguts were fixed in 4% formaldehyde in PBS for 1 h and infiltrated with 15% sucrose/PBS overnight at 4 °C. The medium was changed to 7.5% gelatin containing 15% sucrose at 37 °C for 1 h and the tissues were rapidly frozen at −50 °C in methylbutane (Sigma-Aldrich, Budapest, Hungary). Frozen sections were cut at 12 μm thickness, collected on poly-l-lysine-coated slides (Sigma-Aldrich, Budapest, Hungary) and stained by immunocytochemistry as previously described [60]. Briefly, after rehydration, sections were incubated with primary antibodies for 1 h. Primary antibodies used were anti-p75 ^NTR^ (1:2000, kind gift of Dr. Louis Reichardt [61]) and anti-Versican (1:500, kind gift of Dr. Maria T. Dours-Zimmerman [62]). Sections were incubated with biotinylated goat anti-rabbit IgG (1:200, Vector Laboratories BP-9100-50, Burlingame, CA, USA) for 45 min and avidin-biotinylated peroxidase complex (Vectastain Elite ABC kit, Vector Laboratories PK-6105, Burlingame, CA, USA) for 30 min at room temperature. Endogenous peroxidase activity was quenched by incubation for 10 min with 3% hydrogen peroxide (Sigma-Aldrich, Budapest, Hungary) in PBS. Binding sites of the primary antibodies were visualized by 4-chloro-1-naphtol (Sigma-Aldrich, Budapest, Hungary).

## 3. Results

### 3.1. Polydactyly in the ta^2^ Hindlimb Does Not Correlate with Ectopic Expression of PTCH1

In the developing limb, *Sonic hedgehog* (*SHH*), expressed in a region of the posterior margin called the zone of polarizing activity (ZPA), is essential for the specification of the anterior-posterior (AP) axis [63,64]. The polydactyly phenotype of the *ta*^2^ embryo has been the subject of extensive study [48,52,53,65,66,67,68]. Skeletal analyses revealed that HH39 control hindlimbs possessed four digits and three (digits 1 and 2) to four (digits 3 and 4) phalangeal bones (Figure 1A). Conversely, HH39 *ta*^2^ hindlimbs possessed 7–8 digits composed of only cartilage, and a singular fused metatarsal element (Figure 1B). Previous studies that investigated Hh signaling in the *ta*^2^ limb focused on stages after morphological differences (loss of asymmetry) were first visible [51,52,53]. These studies suggested that polydactyly in the *ta*^2^ was a consequence of constitutive activation of the Hh pathway as *PTCH1* and *GLI1* were ectopically expressed in the anterior portion of the limb bud in the absence of *SHH* expression [52]; however, these analyses were performed only after the *ta*^2^ limb bud had lost its asymmetry and adapted its characteristic paddle shape (Figure 1C). Additionally, these studies were conducted before it was possible to discern between control and heterozygote embryos. Thus, HH24 embryos were genotyped and expression of Hh signaling components were assayed via RNAscope in situ hybridization and quantified in control and *ta*^2^ embryos to re-examine the molecular cause of the *ta*^2^ polydactyly phenotype. Before performing molecular analyses for Hh pathway genes, we confirmed that there was no significant difference in the areas of HH24 control and *ta*^2^ hindlimbs (Appendix A). In control hindlimbs, *SHH* expression was localized to the ZPA in the posterior portion of the hindlimb (Figure 1D). There was no discernable difference in the expression of *SHH* in stage-matched *ta*^2^ embryos (Figure 1E) as the area of the ZPA was not significantly different between control and *ta*^2^ hindlimbs (Appendix A). Conversely, the *PTCH1* expression domain was reduced in the *ta*^2^ hindlimb, when compared to control hindlimbs, and ectopic expression of *PTCH1* was not observed in the anterior portion of the hindlimb (Figure 1F–I). Quantification of *PTCH1* expression via puncta counting verified a significant reduction of *PTCH1* expression in the *ta*^2^ hindlimbs (Figure 1J). Thus, these results suggested that the ciliopathic polydactylous phenotype in the *ta*^2^ limb was not a consequence of ectopic *PTCH1* expression.

### 3.2. Loss of Ventral Neuronal Cell Types in ta^2^ Neural Tube Correlates with Reduced PTCH1 Expression

The vertebrate neural tube is patterned along the dorsal-ventral (DV) axis by graded Shh activity originating from the notochord and subsequently from the ventral floor plate [69,70]. Mutations in *Shh* results in a loss of ventral neural progenitor fates [64]. Previous experiments suggested that Hh-mediated neuronal patterning proceeded normally in the *ta*^2^ neural tube as there were only modest disruptions in neuronal differentiation [54]; however, these experiments were performed at HH25, well after DV patterning and neural progenitor specification was completed [71], and heterozygosity was not considered. To determine if Hh-mediated DV patterning and neural progenitor specification was aberrant, HH20 control and *ta*^2^ embryos were genotyped and expression of neural DV markers and Hh signaling components were assayed via RNAscope in situ hybridization.

As previously reported, *NK2 homeobox 2* (*NKX2.2*), a marker of p3 neuronal progenitors [72], was expressed as a solid stripe in the ventral aspect of the control neural tube (Figure 2A). While the *NKX2.2* domain was maintained in the ventral aspect of the *ta*^2^ neural tube, the boundary of the domain was disrupted, and overall expression appeared less robust (Figure 2B). Expression quantification via puncta counting confirmed that *NKX2.2* expression was significantly reduced in *ta*^2^ embryos (Figure 2C). Next, expression of additional neuron progenitor markers including *Oligodendrocyte lineage transcription factor 2* (*OLIG2*) [73] and *Iriquois homeobox 3* (*IRX3*) [74] were examined. *OLIG2* was expressed in a defined stripe, dorsal to the *NKX2.2* expression domain in control embryos (Figure 2D). In the *ta*^2^ neural tube, the ventral aspect of the *OLIG2* domain lacked a defined boundary (Figure 2E), and puncta quantification confirmed a significant downregulation of *OLIG2* expression (Figure 2F). Lastly, expression of *IRX3*, which was expressed by p0–p2 and dorsal neuronal progenitors, was examined. In control embryos, *IRX3* was expressed in the dorsal half of the neural tube (Figure 2G). In *ta*^2^ embryos, the *IRX3* domain was expanded into the more ventral aspect of the neural tube (Figure 2H). Quantification of *IRX3* puncta confirmed a significant upregulation in *IRX3* expression (Figure 2I). It should be noted that a significant difference in the ratio of gene expression relative to total neural tube area was not detected (Appendix A). Together, these results revealed that DV patterning was indeed disrupted in the neural tube of *ta*^2^ embryos.

Diminished p3 and motor neuron progenitor domains, coupled with the expanded dorsal neuron progenitor domains, suggested that Hh signaling activity was decreased in the *ta*^2^ neural tube. To test this hypothesis, we assayed expression of *SHH* and *PTCH1*. While the expression of *SHH* in the notochord and floor plate of *ta*^2^ embryos was detected (Figure 3A,B), the *SHH* expression domain in the floor plate was significantly reduced when compared to controls (Figure 3G). Concordant with this result, *PTCH1* expression (as per puncta quantification) was also significantly downregulated in the *ta*^2^ neural tube when compared to controls (Figure 3C–F,H), despite the area of *PTCH1* expression remaining equivalent between control and *ta*^2^ embryos (Appendix A). Collectively, these results suggested that aberrant neural tube patterning in the *ta*^2^ was a consequence of decreased Hh pathway activation in the ventral neural tube.

### 3.3. Midfacial Widening in the ta^2^ Correlates with Increased PTCH1 Expression in the Frontonasal Prominence

Early craniofacial patterning requires an inductive interaction from the brain to the face as *SHH* from the forebrain induces *SHH* expression in the ectoderm of the frontonasal prominence (FNP) [75]. *SHH* expression in the FNP abuts with *Fibroblast growth factor 8* (*FGF8*) expression to molecularly determine the frontonasal ectodermal zone (FEZ), a signaling domain required for midfacial development [76,77]. Loss of *SHH* in the FEZ during early craniofacial development results in midfacial collapse, whereas a gain-of-Hh pathway activity results in midfacial widening [36,78]. The *ta*^2^ embryo presents with a shortened and widened upper beak, a phenotype characteristic of a gain-of-Hh signaling phenotype [42,51,75,78]. However, the mechanisms underlying this phenotype have yet to be addressed due to the inability to identify mutants before morphological presentation. Thus, to determine if the formation of the FEZ and Hh signaling were disrupted in the *ta*^2^, HH20 embryos were harvested, genotyped and assayed for expression of molecular markers via RNAscope in situ hybridization. 

To determine if the FEZ was disrupted in *ta*^2^ embryos, *SHH* and *FGF8* expression was examined. Relative to control embryos, *SHH* expression in *ta*^2^ embryos was expanded throughout the FNP ectoderm (Figure 4A,B). Similarly, *FGF8* expression was more dispersed and ventrally expanded in *ta*^2^ FNP ectoderm, relative to control embryos (Figure 4C,D). Accordingly, the FEZ was expanded in *ta*^2^ embryos when compared to stage-matched control embryos (Figure 4E,F). These results suggested the widened and shortened upper beak phenotype in *ta*^2^ embryos could be due to aberrant formation and function of the FEZ.

Since the brain serves as a molecular and structural scaffold necessary for early craniofacial development [79] and FEZ formation is initiated by signals from the neuroectoderm, brain patterning in *ta*^2^ embryos was examined. *FGF8* expression was detected in the forebrain and midbrain/hindbrain boundary (MHB) (Figure 4G) [79,80,81]. In *ta*^2^ embryos, the *FGF8* expression domain was expanded, but diffuse throughout the developing brain, as it was expressed in the forebrain, MHB, and throughout the midbrain and hindbrain neuroepithelium (Figure 4H). *Orthodenticle homeobox 2* (*OTX2*), a transcription factor that specifies midbrain identity, is antagonized by *FGF8* [80,81]. In control embryos, *OTX2* was expressed in the midbrain (Figure 4I); however, in *ta*^2^ embryos, *OTX2* was expressed throughout all three brain vesicles, even in areas where *FGF8* was expressed (Figure 4J). These results suggested that disrupted brain patterning, together with disruption of Hh signaling and FEZ demarcation, contributes to the craniofacial phenotypes present in *ta*^2^ embryos.

Hh signaling is essential for patterning the ventral aspect of the central nervous system, including the forebrain [82]. To determine if Hh signaling was disrupted throughout the developing brain of *ta*^2^ embryos, Hh pathway activity was assessed. *SHH* was localized to the diencephalon (including the caudal aspect of the diencephalon, or tuberculum posterius) and telencephalon (Figure 5A). In addition to the endogenous regions of *SHH* expression present in control embryos, *ta*^2^ embryos had an ectopic domain of *SHH* in the hindbrain (Figure 5B). As expected, *PTCH1* was expressed adjacent to regions of *SHH* expression in control embryos (Figure 5C,E). In *ta*^2^ embryos, ectopic *PTCH1* expression was observed throughout the neuroepithelium of all three lobes brain (Figure 5D,F). Collectively, these results suggested that the craniofacial phenotypes of *ta*^2^ embryos were a consequence of increased Hh activity in the brain.

### 3.4. Hypoplastic Gastrointestinal Tract in the ta^2^ Does Not Correlate with Increased PTCH1 Expression

The gastrointestinal (GI) tract relies on proper Hh activity during development. *SHH* is expressed in the definitive endoderm during early organogenesis and signals to adjacent mesoderm-derived mesenchyme [32,69]. Genetic loss of *Shh* in murine embryos results in shortened GI tracts [83], whereas overexpression of *Shh* results in organ overgrowth [84]. Despite the *ta*^2^ being a long-utilized avian mutant, the *ta*^2^ GI phenotype has not been previously described. The avian GI tract consists of the esophagus, proventriculus, gizzard, small intestine (midgut), ceca, colon (hindgut), and cloaca (Figure 6A). Similar to controls, *ta*^2^ embryos possessed all elements of the GI tract (Figure 6B); however, the GI tracts of *ta*^2^ embryos were significantly shorter than those of stage-matched control embryos (Figure 6C). Measurements of each individual element revealed that the small intestine and colon were significantly smaller in *ta*^2^ embryos when compared to control embryos (Figure 6D,E).

Previous studies demonstrated a role for Hh signaling in enteric neural crest cell (ENCC) migration into the hindgut via modulating deposition of chondroitin sulfate proteoglycans (CSPGs) that inhibit ENCC migration [85]. To determine if ENCC migration was perturbed in the *ta*^2^ hindgut, P75 immunostaining was performed. Increased P75 immunostaining was observed in *ta*^2^ hindguts when compared to controls (Figure 6F,G). ENCC migration was also perturbed in *ta*^2^ midguts as there was increased P75 immunostaining when compared to controls (Appendix A). To determine the cause of this ectopic migration of ENCCs, immunostaining for the CSPG protein VERSICAN was performed. Indeed, VERSICAN deposition was decreased in *ta*^2^ hindguts, correlating with the observed ectopic ENCC migration (Figure 6H,I). These results suggested that there was a loss of Hh signaling in *ta*^2^ hindgut.

While it is well-established that Hh signaling is disrupted in the *ta*^2^ [36,43,51,52,53,54], *SHH* expression and pathway activity in the *ta*^2^ GI tract has yet to be explored. The gross GI hypoplasia and ectopic ENCC migration phenotypes suggested that Hh signaling activity was decreased in *ta*^2^ hindguts. To test this hypothesis, we spatially and temporally examined *SHH* expression and pathway activity in the developing hindgut. RNAscope in situ hybridization for *SHH* on transverse sections of HH29 hindguts revealed *SHH* expression in the hindgut epithelium was comparable between control and *ta*^2^ embryos (Figure 7A,B). Quantification of the hindgut epithelium area revealed no significant difference between control and *ta*^2^ embryos (Appendix A). In control embryos, *PTCH1* was expressed throughout the mesenchyme in a gradient fashion, with the most robust expression closest to the intestinal epithelium and more diffuse expression in the periphery (Figure 7C,E). In *ta*^2^ hindguts, *PTCH1* expression was expanded and uniform throughout the intestinal mesenchyme (Figure 7D,F). Quantitative analysis of *PTCH1* puncta confirmed a significant increase of *PTCH1* expression in the *ta*^2^ hindgut (Figure 7G). These analyses were repeated at two additional developmental timepoints, HH34 and HH39, and similar significant increases in *PTCH1* expression were observed (Appendix A). These results demonstrated that despite presenting with a loss-of-Hh phenotype, *PTCH1* expression was increased throughout the *ta*^2^ GI tract.

## 4. Discussion

This study explored the etiology of both gain- and loss-of-Hh phenotypes in various tissues of the avian ciliopathic model *ta*^2^. The recent identification of the genetic cause of the *ta*^2^ phenotype allowed us to both readdress signaling centers previously studied, as well as characterize those never before explored. These analyses revealed that the phenotypic presentation of some signaling centers (neural tube and craniofacial complex) correlated with Hh activity, as per the transcriptional readout of *PTCH1* expression, while other signaling centers (limb and GI tract) had seemingly contradictory readouts (Figure 8).

In the neural tube, Hh-mediated patterning is initiated through an induction of *SHH* expression in the floor plate from the notochord [69,70]. Similarly, in the craniofacial complex, Hh-mediated patterning of the FEZ relies on induction of *SHH* expression in the FNP ectoderm from *SHH* in the forebrain [75]. The neural tube and the FNP are considered ‘secondary’ Hh domains as their *SHH* expression and Hh-mediated patterning events are contingent upon induction by a ‘primary’ source of Shh ligand such as the notochord or forebrain neuroepithelium [75]. While *SHH* expression is maintained in the ‘primary’ signaling domains, *SHH* expression in the ‘secondary’ Hh domains of the *ta*^2^ was perturbed. These results suggest that primary cilia are required for induction of ‘secondary’ Hh domains and *PTCH1* expression correlates with the overall Hh pathway activity and resulting phenotypes in these tissues. 

The limb and GI tract are not dependent upon induction of a secondary Hh signaling domain, and also rely on the activity of Gli transcription factors for development and patterning. Polydactyly is commonly associated with a gain-of-Hh function, as digits are lost in absence of Hh and the number of digits is increased when Gli3R is lost [86,87,88]. While the data herein is not contradictory to the onset of polydactyly [89], it is distinct from that obtained from embryos after morphological variation was evident [52]. Since the *ta*^2^ mutant is ciliary in nature, it remains most likely that loss of Gli3R activity is the predominant mechanism for the polydactylous phenotype. Additionally, the developing GI tract in the *ta*^2^ embryo presents with phenotypes consistent with a loss-of-Hh function and increased *PTCH1* transcription was observed. As such, *PTCH1* expression in this tissue did not correlate the ultimate readout of pathway activity. Understanding differences in Hh-mediated patterning in each tissue is essential for the development of targeted therapeutic strategies in patients that present with ciliopathies.

While ciliopathic mutations can reveal how a specific tissue transduces the Hh pathway, they can also reveal distinct roles for ciliary proteins in this process [90,91]. An apt example that demonstrates this point is the different phenotypes within Hh-dependent signaling centers between the *ta*^2^ and a long associated ‘sister’ avian mutant called the *talpid^3^* (*ta^3^*). The causative mutation of the *ta^3^* is an insertional mutation in the *KIAA0586* gene that encodes the TALPID3 protein [71], a ciliary centrosomal protein that is required for basal body docking to the cell surface prior to ciliogenesis [92,93]. While the *ta*^2^ and *ta^3^* phenotypes both arise due to mutations in centrosomal proteins, the amount of ciliary extension between the avian mutants is different as approximately 20% of *ta*^2^ cells extend a cilium, whereas there is a complete loss of ciliary extension in *ta^3^* embryos [43,92]. Despite *ta*^2^ and *ta^3^* mutants both possessing similar polydactylous limb phenotypes [94], phenotypic presentations in the neural tube, the craniofacial complex, and the GI tract are distinct between the two mutants. The *ta^3^* has a more severe DV patterning defect (a complete loss of ventral cell types) in the neural tube than the *ta*^2^ [71]. While the *ta*^2^ phenocopies craniofacial phenotypes associated with Oral-facial-digital syndrome subtype 14 (OFD14) [47,95,96,97], the *ta^3^* presents with craniofacial phenotypes associated with Joubert syndrome including hypertelorism and hypoplastic jaws [98,99,100,101,102]. While the *ta*^2^ and *ta^3^* GI tracts are both significantly shorter than control embryos, the intestinal ENS phenotypes are divergent from each other as the *ta*^2^ hindgut has an increased number of ENCCs whereas *ta^3^* guts have reduced numbers of ENCCs [103,104]. The differences in the phenotypic presentations between *ta*^2^ and *ta^3^* embryos demonstrate not only the importance of functional primary cilia for Hh signal transduction, but also the divergent effects mutations in ciliary proteins can have on Hh signaling and embryonic development. 

In addition to avian mutants being utilized to explore the function of ciliary genes like *C2CD3* and *KIAA0586* during development and disease, orthologous mutations have also been identified. The murine *Hearty* mutant, an ENU-induced *C2cd3* mutation, presents with polydactyly, a dorsalized neural tube and decreased *Ptch1* transcription [44]. The *talpid3* murine mutant presents with a loss of ventral neural progenitors, polydactyly, a loss of midline facial structures, and a loss of *Ptch1* expression [105]. Additionally, the *talpid3* zebrafish mutant presents with a loss of ventral interneuron progenitors in the neural tube as well as symmetric pectoral fin buds that are reminiscent of the early development of the polydactylous *ta^3^* avian limbs [106]. The conservation of phenotypes and organ specific Hh pathway readouts between avian, murine and zebrafish mutants suggests these models will be valuable tools in the continued exploration of the etiology of human ciliopathies. 

While we have explored how a *C2CD3* mutation and impaired ciliogenesis perturbs Hh signaling in four distinct Hh-dependent signaling centers, it should be noted that several other tissues that rely on Hh signaling for proper patterning have yet to be examined in this context. For example, Hh signaling is required to properly pattern the dorsal-ventral axis of the somite that subsequently allows for differentiation into the dermatome (presumptive skin), myotome (presumptive muscle), and the sclerotome (presumptive vertebrae) [107,108,109,110]. Interestingly, previous studies reported that the *ta*^2^ embryo presents with mispatterned muscles and a shortened vertebral column [42,48] and human patients that present with OFD14 possess musculoskeletal anomalies [96,97]. Future work with the *ta*^2^ embryo will assess if and how these particular phenotypes arise due to perturbations in cilia-dependent Hh signaling. 

Despite the close association between cilia and Hh signaling, it is unlikely that Hh signaling is the sole molecular mechanism leading to the phenotypic presentations observed in the *ta*^2^ avian ciliopathic mutant. It is well established that the primary cilium has roles in the transduction of other signaling pathways, including the Wnt and Platelet derived growth factor (PDGF) pathways [3]. Additionally, Bone morphogenetic protein (BMP) and Fibroblast growth factor (FGF) pathways have also been implicated in the development of tissues explored in this study [76,77,111,112,113,114,115,116]. Furthermore, these signaling pathways participate in crosstalk to regulate proper development of embryonic tissues [117,118,119]. How other signaling pathways are affected in *ta*^2^ and other ciliopathic mutants will be a focus of our future studies and will result in a more comprehensive understanding of how primary cilia mediate the transduction of several signaling pathways.

## Figures and Tables

**Figure 1 jdb-09-00012-f001:**
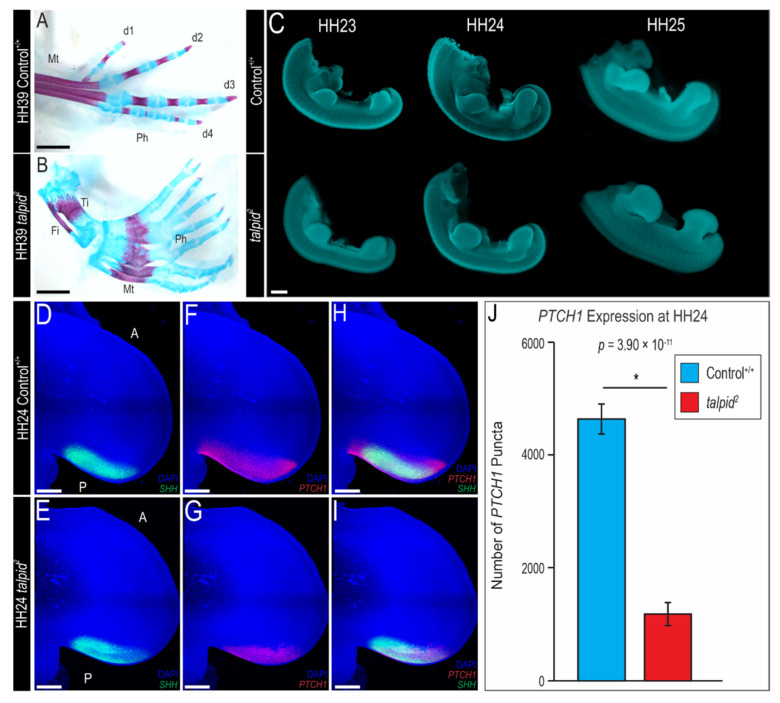
Polydactylous *ta*^2^ phenotype does not correlate with *PTCH1* expression. (**A**,**B**) Alcian blue and Alizarin red staining of a HH39 control^+/+^ (*n* = 5) and *ta*^2^ (*n* = 4) hindlimbs. (**C**) Wholemount HH23-25 control^+/+^ and *ta*^2^ embryos stained with DAPI. (**D**–**I**) RNAscope in situ hybridization for (**D**,**E**) *SHH*, (**F**,**G**) *PTCH1* or (**H**,**I**) both *SHH* and *PTCH1* in HH24 control^+/+^ (*n* = 9) and *ta*^2^ (*n* = 9) hindlimbs. (**J**) Puncta quantification for *PTCH1* expression in HH24 control^+/+^ and *ta*^2^ hindlimbs. A: Anterior, D1–D4: digits, Fi: fibula, Mt: metatarsal element, P: posterior, Ph: phalanges, Ti: tibia. Scale bars: (**A**,**B**) 2 mm, (**C**) 1 mm, (**D**–**I**) 500 µM. Error bars represent the mean data ± s.d. Statistical analysis was performed utilizing Student’s *t*-test (* denotes *p* < 0.05).

**Figure 2 jdb-09-00012-f002:**
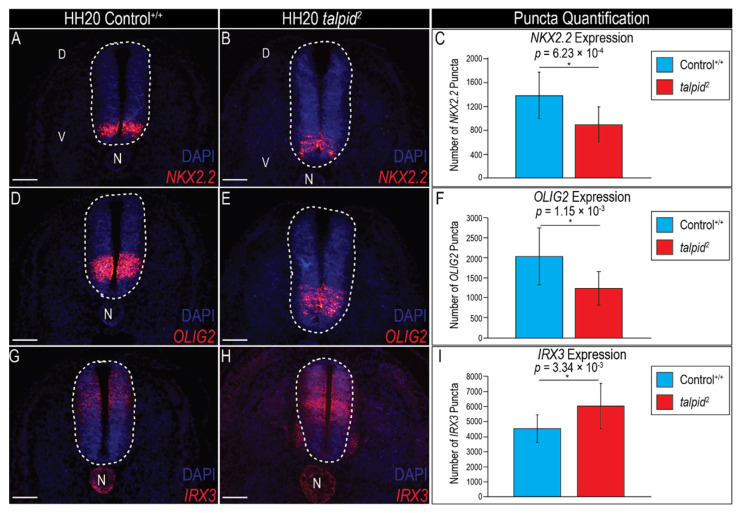
Expression of markers for ventral neuronal progenitors is reduced in the *ta*^2^ neural tube. (**A**,**B**) RNAscope in situ hybridization for *NKX2.2* in HH20 control^+/+^ and *ta*^2^ neural tubes. (**C**) Puncta quantification for *NKX2.2* expression in HH20 control^+/+^ and *ta*^2^ neural tubes. (**D**,**E**) RNAscope in situ hybridization for *OLIG2* in HH20 control^+/+^ and *ta*^2^ neural tubes. (**F**) Puncta quantification for *OLIG2* expression in HH20 control^+/+^ and *ta*^2^ neural tubes. (**G**,**H**) RNAscope in situ hybridization for *IRX3* in HH20 control^+/+^ and *ta*^2^ neural tubes. (**I**) Puncta quantification for *IRX3* expression in HH20 control^+/+^ and *ta*^2^ neural tubes. D: dorsal, N: notochord, V: ventral. White dotted ovals outline the neural tubes. Scale bars: (**A**,**B**; **D**,**E**; **G**,**H**) 100 µm. Error bars represent the mean data ± s.d. Statistical analysis was performed utilizing Student’s *t*-test (* denotes *p* < 0.05).

**Figure 3 jdb-09-00012-f003:**
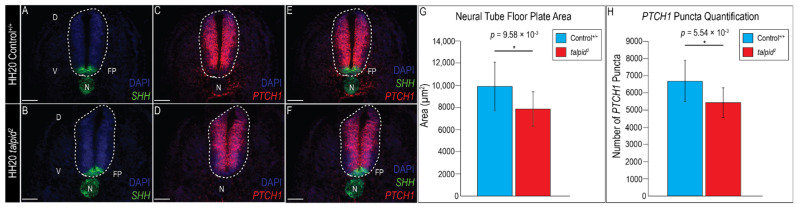
Loss of ventral neural progenitors correlates with decreased *PTCH1* expression. (**A**–**F**) RNAscope in situ hybridization for (**A**,**B**) *SHH*, (**C**,**D**) *PTCH1*, or (**E**,**F**) both *SHH* and *PTCH1* in HH20 control^+/+^ and *ta*^2^ neural tubes. (**G**) Quantification of floor plate area in HH20 control^+/+^ and *ta*^2^ neural tubes. (**H**) Puncta quantification for *PTCH1* expression in HH20 control^+/+^ and *ta*^2^ neural tubes. D: dorsal, FP: floor plate, N: notochord, V: ventral. White dotted ovals outline the neural tubes. Scale bars: (**A**–**F**) 100 µm. Error bars represent the mean data ± s.d. Statistical analysis was performed utilizing Student’s *t*-test (* denotes *p* < 0.05).

**Figure 4 jdb-09-00012-f004:**
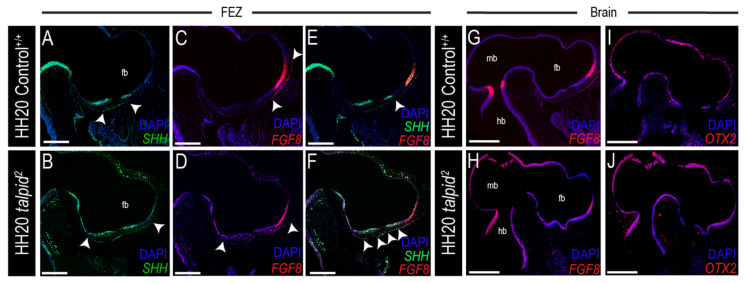
The frontonasal ectodermal zone (FEZ) is expanded and dispersed in *ta*^2^ embryos. (**A**–**F**) RNAscope in situ hybridization for (**A**,**B**) *SHH*, (**C**,**D**) *FGF8* or (**E**,**F**) both *SHH* and *FGF8* in HH20 control^+/+^ and *ta*^2^ craniofacial complexes. Arrowheads in (**A**,**B**) demarcate boundaries of *SHH* expression, arrowheads in (**C**,**D**) demarcate boundaries of *FGF8* expression, and arrowheads in (**E**,**F**) demarcate the FEZ. (**G**,**H**) RNAscope in situ hybridization for *FGF8* in HH20 control^+/+^ and *ta*^2^ brains. (**I**,**J**) RNAscope in situ hybridization for *OTX2* in HH20 control^+/+^ and *ta*^2^ brains. fb: forebrain, hb: hindbrain, mb: midbrain. Scale bars: (**A**–**F**) 250 μm, (**G**–**J**) 500 µm.

**Figure 5 jdb-09-00012-f005:**
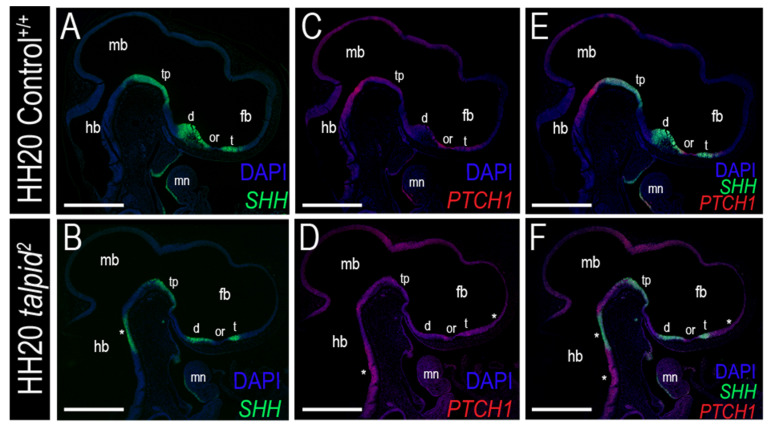
Expanded FEZ correlates with increased *PTCH1* expression. (**A**–**F**) RNAscope in situ hybridization for (**A**,**B**) *SHH*, (**C**,**D**) *PTCH1* or (**E**,**F**) both *SHH* and *PTCH1* in HH20 control^+/+^ and *ta*^2^ brain and craniofacial complex. Asterisks (*) denote ectopic expression of (**B**) *SHH*, (**D**) *PTCH1*, or (**F**) both *SHH* and *PTCH1* in the developing *ta*^2^ brain and craniofacial complex. d: diencephalon, fb: forebrain, hb: hindbrain, mb: midbrain, mn: mandibular arch, or: optic recess, t: telencephalon, tp: tuberculum posterius. Scale bars: (**A**–**F**) 500 µm.

**Figure 6 jdb-09-00012-f006:**
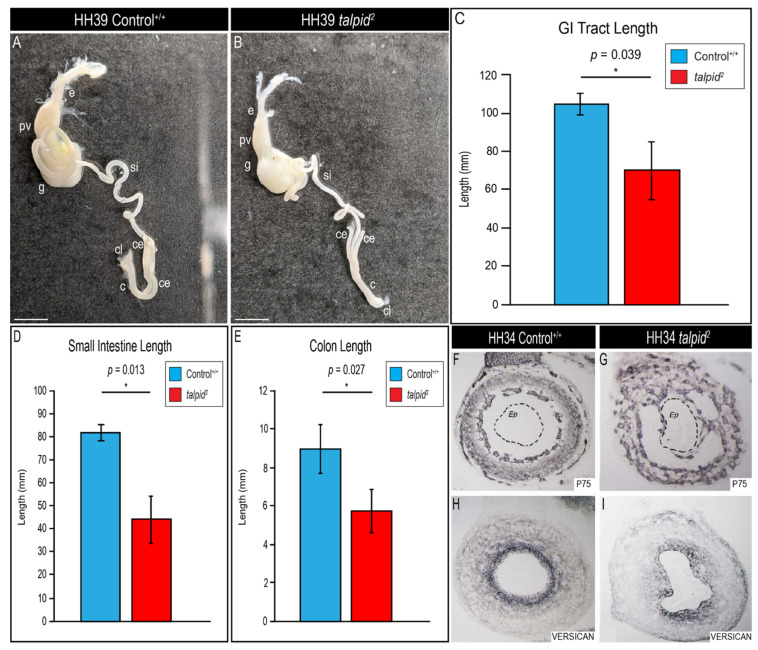
*ta*^2^ embryos present with shortened gastrointestinal (GI) tracts accompanied with ectopic enteric neural crest cell (ENCC) migration. (**A**,**B**) Dissected gastrointestinal tracts from HH39 control^+/+^ (**A**, *n* = 3) and *ta*^2^ (**B**, *n* = 3) embryos. (**C**–**E**) Length measurements of HH39 control^+/+^ and *ta*^2^ (**C**) gastrointestinal tracts, (**D**) small intestines, and (**E**) colons. (**F**–**I**) Immunostaining for (**F**,**G**) P75 and (**H**,**I**) VERSICAN in transverse sections of HH34 control^+/+^ and *ta*^2^ hindguts. Dotted black circles in (**F**,**G**) indicate the intestinal epithelium. c: colon, ce: cecum, cl: cloaca, e: esophagus, Ep: epithelium, g: gizzard, pv: proventriculus, si: small intestine. Scale bars: (**A**,**B**) 6 mm. Error bars represent the mean data ± s.d. Statistical analysis was performed utilizing Student’s *t*-test (* denotes *p* < 0.05).

**Figure 7 jdb-09-00012-f007:**
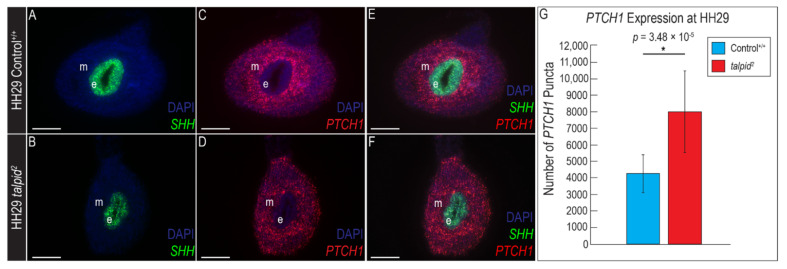
Hypoplastic GI phenotype does not correlate with increased *PTCH1* expression. (**A**–**F**) RNAscope in situ hybridization for (**A**,**B**) *SHH*, (**C**,**D**) *PTCH1* or (**E**,**F**) both *SHH* and *PTCH1* in the hindgut epithelium of transverse sections of HH29 control^+/+^ and *ta*^2^ hindguts. (**G**) Puncta quantification for *PTCH1* in HH29 control^+/+^ and *ta*^2^ hindguts. e: epithelium, m: mesenchyme. Scale bars: 100 µm (**A**–**F**). Error bars represent the mean data ± s.d. Statistical analysis was performed utilizing Student’s *t*-test (* denotes *p* < 0.05).

**Figure 8 jdb-09-00012-f008:**
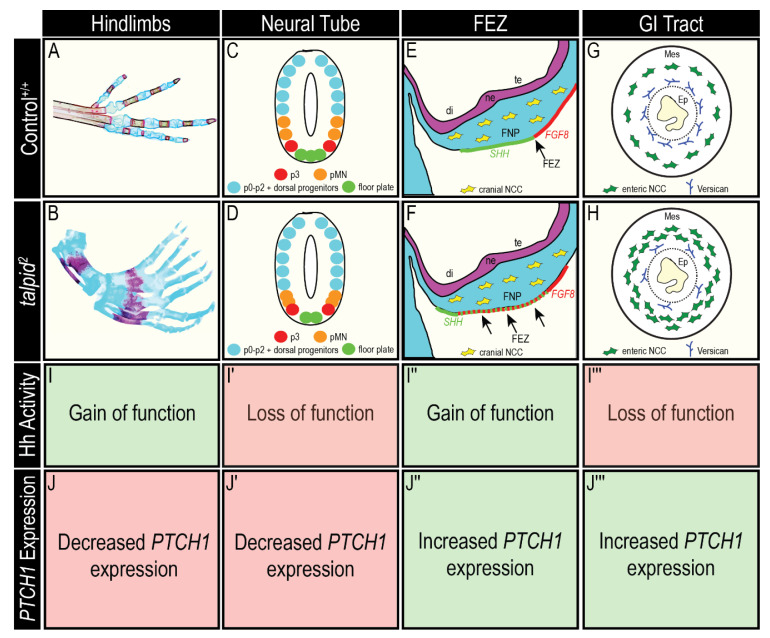
Summary of phenotypes, associated Hh activity, and *PTCH1* expression across Hh-dependent signaling centers in control and *ta*^2^ embryos. (**A**–**H**) Schematics of (**A**,**B**) hindlimb, (**C**,**D**) neural tube, (**E**,**F**) FEZ, and (**G**,**H**) GI tract phenotypes in (**A**,**C**,**E**,**G**) control^+/+^ and (**B**,**D**,**F**,**H**) *ta*^2^ embryos. (**I**–**I**’’’) Level of Hh activity associated with *ta*^2^ phenotypic presentations. (**J**–**J**’’’) Summary of *PTCH1* expression in *ta*^2^ organs. di: diencephalon, Ep: epithelium, FEZ: frontonasal ectodermal zone, FNP: frontonasal prominence, Mes: intestinal mesenchyme, NCC: neural crest cell, ne: neuroepithelium, pMN: motor neuron progenitors, te: telencephalon.

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
