# Peer review of "Mutation in the Ciliary Protein C2CD3 Reveals Organ-Specific Mechanisms of Hedgehog Signal Transduction in Avian Embryos"

_jdb, 2021, doi:10.3390/jdb9020012_

Round 1
Reviewer 1 Report
This is a nice study by the Brugmann group looking at the ta2 mutant in chick. They compare ta2 and control chick embryos and characterise the phenotype in several HH-dependent areas of the organise. This paper fills some of the gaps on what is known about the ta2 mutant and nicely utilises RNAscope technology rather than just colorimetric ISH. Overall it is well written and logically explained. I think the manuscript could benefit with further clarification on a few points listed below.
- It is possible to summarise the known ciliopathy mutations? The authors mention the Ta3 mutant in the discussion but maybe include the mouse/fish mutants as a comparison? This would allow a nice comprehensive view of what has been observed as gain or loss of HH function and at what point the mutations occur for the cilium.
- The authors study key HH-dependent tissues during chick development (limb, NT, cranio-facial and GI). I was wondering whether they could comment on other HH-dependent tissues such as somite development? There are lots of papers by other groups that study HH in sclerotome and myotome formation (see Munsterberg, Borycki, Kalchiem papers for example).
- For Fig 1 could the authors possible examine the area size of the control limbs v ta2 mutant limbs? Also rather than staging, can the authors comment whether ta2 mutants develop at the same speed as controls? This eludes to the fact that HH signalling isn't just about cell specification but has a role in timing as well (see Briscoe papers for example)
- For Fig 1B is it possible to distinguish the digit identity in the ta2 mutant?
- For Figs 2 and 3, is it possible to measure the ratio of expression compared to size of the NT? This analysis could help answer the question on growth/timing of the NT
- In Ta2 mutants, do the authors observe any defects in cell polarity or orientation of the tissues they examine?
- Have the authors investigated whether the number of NCCs are affected in the Ta2 mutants?
- Although the cilium has been greatly investigated for HH signalling and the authors briefly mention other signalling pathways that could be involved, I would like the authors to expand on these in the discussion. Especially when it is known the WNT has a role in DV patterning in the NT so is it possible for HH and other signalling pathways to cross-talk (such a Gsk3b for example) and whether the phenotype is therefore result of potentially other pathways involved such as WNT?
Reviewer 2 Report
The manuscript by Brooks et al. describes phenotypes of ta2 chick embryos. RNAscope was used to identify alterations in quantity and in situ pattern of transcripts, especially those regulated by Hh pathway signaling. Some of the developmental systems studied in the paper have been studied in these mutants before, but generally at later stages of development and with lower precision.
The results are in line with what would be anticipated for a ciliopathy gene, but the data are clear and of high quality, and they should be of value to the community studying these disorders. My comments relate to some of the interpretations provided, which seem either inconsistent with how some of these systems are regulated or are simply confusing to this reader.
- On line 389, referring to induction of secondary Hh domains like the floor plate, the authors state that “these inductive events act at the level of Hh/Ptch”. What does this mean? All ligand-dependent Hh pathway signaling acts at the level of Hh/Ptch, even in zones where the phenotypes occur exclusively via repression of Gli3R formation and are associated with only low level Ptch1 expression (e.g., the more dorsal portions of the ventral neural tube, the more anterior portion of the Hh-sensitive region of the limb bud). On line 392, it is stated that “Alternatively, several tissues rely on patterning at the level of the Gli transcription factors”. Again, all forms of pathway activity rely on such patterning, whether it is in areas lacking ligand (e.g., where patterning requires the presence of Gli3R) or areas with highest ligand levels (which may require Gli2A, like the floor plate, which by the authors’ statements would be in the first group of tissues). It is not clear to me what the authors are trying to say with these statements. I believe the following comment is also relevant to this.
- Figure 8 and lines 200-202: in interpreting the polydactyly phenotype, the authors conclude that the limb/digit phenotype is a gain-of-function with a contradictory readout of decreased Ptch1 expression. But these observations are consistent with failure to form Gli3R in the anterior of the limb bud (driving extra digit formation), while also failing to form high levels of Gli2A in the ZPA (diminishing Ptch1 expression). This is not so much contradictory as indicative of the known requirement of the Hh pathway for cilia to both relieve constitutive Gli3-based repression and activate Gli2-dependent transcription. Lines 421-424 emphasize these “divergent” effects, and I believe that overlooks our current level of understanding of how the Hh/Gli system signals to promote patterning. In fact, the limb phenotype is less a “gain of function” of the Hh pathway, than a loss of Gli3 function.
Minor point:
- Line 48-49: Smo does not accumulate in ciliary tips (Gli2 and Gli3 do). Smo may not be uniformly distributed in cilia, but it is generally seen throughout the ciliary shaft in Hh-stimulated cells.
Author Response
REV2:
The manuscript by Brooks et al. describes phenotypes of ta2 chick embryos. RNAscope was used to identify alterations in quantity and in situ pattern of transcripts, especially those regulated by Hh pathway signaling. Some of the developmental systems studied in the paper have been studied in these mutants before, but generally at later stages of development and with lower precision.
The results are in line with what would be anticipated for a ciliopathy gene, but the data are clear and of high quality, and they should be of value to the community studying these disorders. My comments relate to some of the interpretations provided, which seem either inconsistent with how some of these systems are regulated or are simply confusing to this reader.
- On line 389, referring to induction of secondary Hh domains like the floor plate, the authors state that “these inductive events act at the level of Hh/Ptch”. What does this mean? All ligand-dependent Hh pathway signaling acts at the level of Hh/Ptch, even in zones where the phenotypes occur exclusively via repression of Gli3R formation and are associated with only low level Ptch1 expression (e.g., the more dorsal portions of the ventral neural tube, the more anterior portion of the Hh-sensitive region of the limb bud). On line 392, it is stated that “Alternatively, several tissues rely on patterning at the level of the Gli transcription factors”. Again, all forms of pathway activity rely on such patterning, whether it is in areas lacking ligand (e.g., where patterning requires the presence of Gli3R) or areas with highest ligand levels (which may require Gli2A, like the floor plate, which by the authors’ statements would be in the first group of tissues). It is not clear to me what the authors are trying to say with these statements. I believe the following comment is also relevant to this.
We thank the Reviewer for these comments. We have updated the text to avoid confusion regarding “levels” of Hh pathway activity being active in different tissues (see pg. 12-13 lines 397-408).
What we were referring to is the culmination of genetic studies that suggest that some ciliary proteins (e.g., IFT proteins) act at the heart of the Hh pathway, downstream of the Ptch and Smo, and upstream of Gli transcription factors (Goetz and Anderson, 2011; Huangfu et al., 2003; Huangfu et al., 2005). While our attempt to fit C2cd3 into a similar schema was confusing and not convincing due to a lack of genetic data, we believe it is still likely that disruptions in Hh signaling in ciliary mutants are due to disrupted Gli processing and subsequently impaired pathway feedback.
- Figure 8 and lines 200-202: in interpreting the polydactyly phenotype, the authors conclude that the limb/digit phenotype is a gain-of-function with a contradictory readout of decreased Ptch1 expression. But these observations are consistent with failure to form Gli3R in the anterior of the limb bud (driving extra digit formation), while also failing to form high levels of Gli2A in the ZPA (diminishing Ptch1 expression). This is not so much contradictory as indicative of the known requirement of the Hh pathway for cilia to both relieve constitutive Gli3-based repression and activate Gli2-dependent transcription. Lines 421-424 emphasize these “divergent” effects, and I believe that overlooks our current level of understanding of how the Hh/Gli system signals to promote patterning. In fact, the limb phenotype is less a “gain of function” of the Hh pathway, than a loss of Gli3 function.
We thank the Reviewer for this comment and have edited the text to be clearer. (see pg. 12-13 lines 398-403). For a number of reasons, polydactyly is often referred to as a gain-of-Hedgehog (ie. Lack of digits in Shh mutants, increased number of digits when the Gli3 repressor is lost, and increased digits in ZPA transplants). While decreased PTCH1 expression, in and of itself, is not contradictory with polydactlylous phenotype, it is contradictory with previous reports in the talpid2 of increased PTCH1 expression (Caruccio et al., 1999).
Minor point:
- Line 48-49: Smo does not accumulate in ciliary tips (Gli2 and Gli3 do). Smo may not be uniformly distributed in cilia, but it is generally seen throughout the ciliary shaft in Hh-stimulated cells.
We thank the Reviewer for this comment. We have corrected this on pg. 2, lines 48-49 to state that Smo accumulates within the ciliary axoneme.